# Pore Structure and Gas Content Characteristics of Lower Jurassic Continental Shale Reservoirs in Northeast Sichuan, China

**DOI:** 10.3390/nano13040779

**Published:** 2023-02-20

**Authors:** Tao Jiang, Zhijun Jin, Hengyuan Qiu, Xuanhua Chen, Yuanhao Zhang, Zhanfei Su

**Affiliations:** 1State Key Laboratory of Petroleum Resources and Prospecting, China University of Petroleum, Beijing 102249, China; 2Chinese Academy of Geological Sciences, Beijing 100037, China; 3State Key Laboratory of Shale Oil and Gas Enrichment Mechanisms and Effective Development, Beijing 100083, China; 4Petroleum Exploration and Production Research Institute, SINOPEC, Beijing 100083, China; 5Institute of Energy, Peking University, Beijing 100871, China; 6Unconventional Natural Gas Research Institute, China University of Petroleum, Beijing 102249, China; 7Oil Production Engineering Research Institute, Daqing Oilfield Company, Daqing 163000, China

**Keywords:** Northeastern Sichuan Basin, continental shale gas, pore structure, gas-bearing property

## Abstract

The Jurassic shale in the northeastern Sichuan Basin is one of the main target intervals for continental shale gas exploitation. Research on the pore structure and gas-bearing properties of shales is the key issue in target interval optimization. Through core observation, geochemistry, bulk minerals, scanning electron microscopy, nitrogen adsorption, and isothermal adsorption experiments, various lithofacies with different pore structure characteristics were clarified. In addition, the factors that control gas-bearing properties were discussed, and a continental shale gas enrichment model was finally established. The results show that the Jurassic continental shale in the northeastern Sichuan Basin can be classified into six lithofacies. Organic pores, intergranular pores, interlayer pores in clay minerals, intercrystalline pores in pyrite framboids, and dissolution pores can be observed in shale samples. Pore structures varied in different shale lithofacies. The contact angle of shales is commonly less than 45°, leading to complex wettability of pores in the shales. Free gas content is mainly controlled by the organic matter (OM) content and the brittleness in the Jurassic shale. The adsorbed gas content is mainly controlled by the OM content, clay mineral type, and water saturation of the shales. The enrichment mode of the Lower Jurassic continental shale gas in the northeastern Sichuan Basin is established. Paleoenvironments control the formation of organic-rich shales in the center part of lakes. The “baffle” layer helps the confinement and high pressure, and the complex syncline controls the preservation, forming the enrichment pattern of the complex syncline-central baffle layer.

## 1. Introduction

In recent years, great progress has been made in both the theory and practice of shale gas exploration and development in China, which has also experienced a golden and rapid development [1,2,3,4]. Continental shale has a wide distribution, multiple layers, abundant geological resources, and great potential for exploration and development. Good oil and gas exist in shales, and a certain amount of oil and gas production has been achieved [5,6]. Continental shale distribution, affected by depositional environment, paleo-water depth, and paleoclimate, has the characteristics of thin single-layers, large thickness, rapid facies change, complex lithologic assemblages, various types of OM, large variation in maturity, and diverse pore structures [7,8].

The hydrocarbon generation capacity, reservoir capacity, and tectonics construction-reconstruction capacity of shale are all important factors affecting the accumulation and distribution of shale gas [9,10]. A complex pore-fracture system is formed due to the structural burial depth, diagenetic evolution, and hydrocarbon generation-expulsion. These pore-fracture systems are both gas storage spaces and seepage channels and are also the main factors affecting the storage capacity and production of shale gas [5,11]. There are many factors affecting the pore structure of shale, and researchers at home and abroad believe that it has a corresponding relationship with the total organic carbon (*TOC*) content of shale, the content and distribution of minerals, the degree of diagenesis, and the thermal evolution of OM (OM). According to the classification of the genesis [12], matrix pores include four types: residual primary pores, unstable mineral dissolution pores, clay mineral interlayer pores, and OM pores. Among them, clay mineral interlayer pores and organic pores are primary in continental shales [13]. This is a significant difference between shale reservoirs and conventional sandstone reservoirs, and it also provides a model basis for the quantitative characterization of reservoir space. Matrix pores can be further classified into micropores in brittle minerals, micropores in OM, and micropores in clay minerals [14,15,16].

The natural gas in shale reservoirs mainly exists in the adsorbed state and the free state, and the dissolved state is rare [17]. Due to the existence of nanoscale particles such as OM and clay minerals in shale, a large specific surface area is formed. Natural gas is mostly adsorbed on the surface of these substances in the form of molecules through physical and chemical action or in the form of filling in the reservoir within the micropores. In the process of formation subsidence and uplift, the natural fractures and matrix pores formed by the structure and diagenesis are the main places for the occurrence of free natural gas, and the production of free natural gas accounts for more than 50% of the total natural gas production [18,19]. However, kerogen in shale and kerogen pyrolysis products such as asphaltenes and crude oil can be used as liquid substances for natural gas to dissolve. The content of dissolved natural gas only accounts for 0.1% of the total natural gas, and the proportion is due to changes in solubility [20].

Several sets of high-quality shale formations have developed in the Jurassic strata in the northeastern Sichuan Basin, which is an important interval for increasing shale gas reserves and production in China [21,22]. There are three sets of deep to semi-deep lacustrine mud shale, namely the Dongyuemiao Member, the Da’anzhai Member, and the Lianggaoshan Formation [23,24]. Since 2009, more than 20 exploratory wells have been tested for Jurassic continental shale reservoirs, and many wells have obtained medium to high-production oil and gas flows. For example, Well YB21 obtained 50.7 × 10^4^ m^3^/d of high-yield shale gas flow in the Da’anzhai member, and Well YB9 obtained 16.6 t/d of shale oil and 1.22 × 10^4^ m^3^/d of gas. Well FY10HF obtained daily gas production of 5.57 × 10^4^ m^3^ and oil of 17.6 m^3^ in the mudstone test of the Dongyuemiao Member; TY1HF obtained daily gas production of 7.5 × 10^4^ m^3^ and oil of 9.8 m^3^ in the Lianggaoshan Formation mudstone test, making a major breakthrough. However, the production and gas-bearing properties of different wells are quite different. Many studies confirmed that shale pore structure had an important impact on shale gas enrichment [25,26]. Currently, the microscopic reservoir characteristics of Jurassic continental shale in the northeastern Sichuan Basin are not well understood, and the controlling factors for gas-bearing properties of the reservoir are not clear, which restricts the in-depth understanding of the enrichment mechanism of continental shale gas [27,28]. This study selected the continental shale of the Da’anzhai Member and Dongyuemiao Member of the Lower Jurassic Ziliujing Formation in the northeastern Sichuan Basin as the research objects. By analyzing the geological and geochemical characteristics of continental shale, reservoir pore structure, and gas-bearing properties, it is hoped to determine the main pore types of shale and the influencing factors on gas-bearing properties. The results are of great significance for the gas-bearing evaluation of continental shale gas, prediction of favorable areas, and promotion of Jurassic shale gas exploration in the Sichuan Basin.

## 2. Materials and Methods

### 2.1. Geological Setting and Samples

The peripheral orogenic belt that developed in the Jurassic resulted in the later late Triassic lacustrine that connected the ocean and gradually disappeared, the Sichuan Basin was enclosed by the lower Yangtze plate, and the central area was transformed into an inland lake [29]. The Lower Jurassic Ziliujing Formation developed a succession of thick dark mud shale, which is one of the main provenance rock series in the entire basin and one of the main shale gas production series [30,31]. This set of mud shale formations generally developed in the whole basin has become a set of regional caprocks well sealed. Integrated with the previous classification scheme, the Ziliujing Formation is divided into the Zhenzhuchong Member, Dongyuemiao Member, Ma’anshan Member, and Da’anzhai Member from bottom to top [32,33] (Figure 1). Among them, the Dongyuemiao and Da’anzhai members are the main exploration intervals for continental shale gas and are composed of dark shale interlayered with shell limestone or sandstone. The dark shale in the Lower Jurassic Ziliujing Formation is generally well-developed, with an average thickness of approximately 60 m. The stratum is the thickest in the northeast of the basin, and the thickness of the shale gradually decreases from the northeast to the southwest of the basin. The dark shale deposits in Wanzhou and Liangping areas in the northeastern part of the basin are the thickest, reaching 260 m; the thickest shale deposits in Cangxi and southern areas of the basin can be up to 240 m. The dark mud shale in the Da’anzhai Member of the Ziliujing Formation of the Lower Jurassic is well developed locally. Thick mudstone layers are developed in the Yingshan, southern, and Pingchang areas in the northern basin, with a thickness of more than 55 m and a relatively uniform distribution. In company with the Dongyuemiao Member, they form the main shale gas interval of the Ziliujing Group. Twenty shale reservoir samples were taken from Well YL4 in northern Sichuan, and this study takes Well XL101 in eastern Sichuan as the research object. The sample number, depth, and geological parameters are shown in Table 1 and Figure 2.

### 2.2. Analytical Methods and Sampling

The examined 10 shale samples (No.1–No.10) from the Jurassic Dongyuemiao Member were collected from the Well XL-101 in the eastern Fuling area. The other examined 10 shale samples (No.11–No.20) from the Da’anzhai Member were collected from the Well YL-4 in the Yuanba area (well locations are shown in Figure 1). The detailed experiment procedures were described below.

#### 2.2.1. X-ray Diffraction (XRD)

All samples were analyzed with X-ray diffractometry (XRD). The examined shale samples were first washed with ethanol to remove possible oil contaminants, then dried at temperatures below 60 °C for 24 h. Two-gram samples were crushed with a prototype crusher and then submerged in an agate mortar to less than 40 microns. The test piece was made by using the back pressure method, and the diffraction intensity of the test surface was measured by the instrument. The baseline was reasonably selected for qualitative and quantitative analysis. The testing procedure used was the SY/T 5163-2010 Standard.

#### 2.2.2. Scanning Electron Microscopy (SEM)

Argon ion polishing scanning electron microscopy was conducted at the State Key Laboratory of Shale Oil and Gas Enrichment and Effective Development of SINOPEC, Wuxi, Jiangsu Province. Shale samples were cut into appropriately sized blocks along the vertical bedding direction, and then a German Leica TXP fine grinder mechanically flatted the shale surface. Appropriate working parameters were set, and the sample surface was bombarded with argon ions using the German Leica RES102 ion thinner. In this experiment, the working voltage of the argon ion polishing instrument was used under operating conditions of 5 KV and 2.2 mA, and the polishing time was 3–4 h. The polished sample was fixed on the sample table with conductive adhesive. A layer of gold film (10–20 nm) was plated on the surface to increase the electrical conductivity and to inhibit sample charging before observations with the FEI Helios 650. Additionally, the secondary electron (SE) and the backscattered electron (BSE) were detected with an accelerating voltage of 2.0 KV, a working distance of 3–4 mm, and an electron beam current of 50 pA–0.2 nA. The different contrast and morphology features were used to empirically distinguish minerals and pores, and the identification of minerals was carried out by X-ray energy dispersive spectrometry (EDS) measurements.

#### 2.2.3. Low-Pressure N_2_ Adsorption and High-Pressure Mercury Experiments

The sample was analyzed using a JW-BK222-type instrument, manufactured by the Cnpowder Company of China. The instrument had a detection limit of 0.0005 m^2^/g in specific surface area and 0.0001 mL/g in pore volume. The pressure control interval was less than 0.1 KPa by the application of the balanced pressure intelligent control method, and the maximum pressure point of adsorption can be automatically brought under control. A 1–2 g sample was smashed to 60–80 mesh. A vacuum was applied at 110 °C for 14 h, and then a nitrogen isothermal adsorption-desorption experiment was performed under the condition of a liquid nitrogen atmosphere (77.4 K), the procedures referenced the Chinese national standard GBT21650.2-2008. A multipoint BET model was carried out to calculate the specific surface area of the pore size distribution, which was received by adsorption curves by the BJH model. The MICP was implemented by making use of an AutoPoreIV9520 capillary pressure curve determinative instrument made by MICROMERITICSTM. The testing range was 3 nm–1000 μm, and the precision was less than ±0.0001 mL. Samples that were tested by NMR were evaluated by AutoPoreIV9520 in consistency with Chinese national standard GBT21650.1-2008 after being dried at 60 °C for 48 h.

#### 2.2.4. Contact Angle Experiment

Contact angle measurements were performed at China University of Petroleum Beijing using an LSA200 video optical contact angle tensiometer. The wetting process is related to the interfacial tension of the system. When a drop of liquid falls on a horizontal solid surface, when equilibrium is reached, the resulting contact angle and each interfacial tension conform to the following Young equation: If the surface contact angle θ < 90°, the solid surface is hydrophilic, that is, the liquid more easily wets the solid, and the smaller the angle is, the better the wettability. If θ > 90°, the solid surface is hydrophobic, i.e., the liquid does not easily wet the solid, and it easily moves on the surface. The principle of the shape image analysis method is to drop a droplet on the surface of a solid sample, obtain the shape image of the droplet through a microscope and a camera, and then use digital image processing and some algorithms to calculate the contact angle of the droplet in the image.

## 3. Results

### 3.1. Mineral Composition Characteristics of the Jurassic Continental Shales

The results show that the Dongyuemiao shales and the Da’anzhai shales are rich in clay minerals, followed by quartz and calcite, and a few feldspar, dolomite, pyrite, and anhydrite minerals. The content of clay minerals ranges from 33.3% to 67.4%, with an average of 52.8%. Brittle minerals are mainly composed of quartz, calcite, dolomite, and feldspar, with an average of 34.8%. Quartz content is between 20.1% and 55.0%, with an average of 30.6%. Calcite is locally enriched, with a range of 0.5–32.1% (11.3% on average). Dolomite content is between 0 and 3.8%, with an average of 0.4%. The feldspar content ranges from 0 to 9.5%, with an average of 2.0%. The clay minerals were mainly composed of mixed layers, with an average value of 69.0%. The illite content was between 7.0% and 22.0%, with an average content of 15.0%, and the contents of chlorite and kaolinite are relatively low, with average contents of 9.0% and 7.0%, respectively. Various mineral components in shale affect the development of reservoir pores, gas occurrence phases, preservation conditions, and fracturing reproducibility in later stages [5,11,34].

### 3.2. Geochemical Characteristics of Continental Jurassic Shale

The total organic carbon (*TOC*) content of shale samples from the Dongyuemiao Member and Da’anzhai Member in the northeastern Sichuan Basin was analyzed. The *TOC* of the Da’anzhai member in the Yuanba area is between 0.70% and 2.49%, with an average of 1.25%; the *TOC* of the Dongyuemiao member in the northern part of the Fuling area was between 0.58% and 2.06%, with an average of 1.53%. Four samples are organic-rich (*TOC* > 2%), accounting for 20.0% of the total number of samples. Most of the samples have *TOC* content below 2%. Nine samples have a *TOC* content within the range of 1–2%, which can be regarded as more favorable shale development intervals.

The vitrinite reflectance (*R*_o_) was used as the evaluation index of the shale maturity in the study area. A total of 20 shale core samples from the Da’anzhai and Dongyuemiao members were tested. The *R*_o_ range of the shale samples in the study area was between 0.96% and 1.44%, with an average value of 1.17%. All samples have the maturity over oil window, ranging from the end of oil generation to the early stage of gas generation. The thermal evolution degree of OM has high values in the Tongjiang–Nanjiang–Bazhong area in the northwest of the study area and the Wanzhou area in the east of the study area, and the *R*_o_ was generally above 1.4% in the above areas.

The vitrinite content in the samples from the artesian well group was the highest, ranging from 42.7% to 92.8%, with an average value of 66.8%, followed by the inertin group, with a content ranging from 3.4% to 55.7%, with an average value of 28.4%. The cytoplasmic content was relatively small, with an average of 1.7%. Through TI calculation, it was found that the main types of kerogen in continental shale are mainly type II_2_ and type III, which also indicated that the parent material is mainly derived from terrigenous higher plant debris.

### 3.3. Lithofacies Classification and Characteristics of Continental Jurassic Shale

According to different mineral compositions (clay minerals, carbonate minerals, and siliceous minerals), the three-end member method is used to make triangle diagrams, and four types of shale lithofacies are divided into four types of shale facies with 50% content as the boundary. The four types of shale facies are clay shale facies (Type I), siliceous shale facies (Type II), calcareous shale facies (Type Ⅲ), and mixed shale facies (Type IV) [35,36] (Figure 3). Clay shale facies, siliceous shale facies, and mixed shale facies are developed in the Da’anzhai Member and Dongyuemiao Member in the northeastern Sichuan Basin. In addition, combined with *TOC* content, shale facies can also be divided into organic-rich shale lithofacies, medium organic shale lithofacies, and organic-poor shale lithofacies. The shale with high *TOC* and high gas content mainly develops mud-rich siliceous shale facies, mud-rich/silicon mixed shale facies, and siliceous shale facies. Combined with the observation of thin sections, the lithofacies of continental shale are further refined according to different structural characteristics. A horizontal sedimentary layer greater than 1 mm is defined as a layered structure, a layered structure is defined as less than 1 mm, and a block structure was defined as have no obvious bedding [37,38]. Therefore, up to 36 shale facies exist based on total organic carbon content, bedding structure, and mineral composition. Six shale lithofacies are mainly developed in the Lower Jurassic Ziliujing Formation in the northeastern Sichuan Basin, which are organic-rich lamellar clay shale; organic-rich layered clay shale; medium organic layered mixed shale; medium organic layered silicon shale; organic-poor lamellar clay shale; and organic-poor massive mixed shale.

### 3.4. Pore Development Types of the Jurassic Continental Shales

As oil and gas storage spaces, pores in shale have also received extensive attention [39,40,41,42]. Shale is an extremely dense reservoir with pore sizes ranging from nanometres to micrometers. Shale reservoirs have the ability to store adsorbed gas and free gas and seepage ability, which are directly related to the pore structure [42,43,44,45]. According to the classification of genesis, matrix pores include four types: residual primary pores, unstable mineral dissolution pores, clay mineral interlayer pores, and OM pores, and their genesis, main characteristics, and development degree vary greatly [14,15,16]. Different organic microscopic components in the study area have different pore-forming abilities. The vitrinites and silks formed by the transformation of the lignocellulosic tissues of higher plants basically do not develop organic pores, while hydrogen-rich vitrinites and organic pores developed to different degrees in solid asphalt (Figure 4a–c). The OM pores are mostly nanoscale, and the pore size is mainly between 3 and 30 nm. The mesopores contribute more to the specific surface area and pore volume of the shale in the study area, so they play a more positive role in the accumulation of shale gas.

Micropores between illites are the main type of pores between clay minerals. Montmorillonite transforms into illite with increasing burial depth. During the transformation, the volume of minerals decreases, and microcracks (pores) are generated [46,47]. The average content of clay minerals in the two key layers, Dongyuemiao and Da’anzhai, is close to 50%. Pores are mainly distributed between the illite sheets and between the illite and mica sheets, and the shapes are mostly slit-shaped, triangular, and polygonal (Figure 4d–f). Controlled by the difference in *TOC* and pore development degree of different samples, the pores are filled with asphalt to different degrees. With the continuous increase in burial depth, the porosity of early clay mineral pores decreases rapidly under the action of strong compaction.

Intergranular pores are one of the major types of pores in the shale in this study. They are the primary pores remaining between the particles that experienced the arrangement and accumulation of diverse mineral particles and diagenetic transformation [48,49]. Our study shows that the intergranular pores of the shale in the Ziliujing Formation are mainly formed by the contact of brittle grains and plastic grains, and their shapes are mainly triangular, polygonal, elongated, and irregular (Figure 4g,h). Their pore size is from nanometres to micrometers, and the range is wide. The remaining intergranular pores that are not filled with asphalt can be preserved owing to the random accumulation of clay minerals and rigid particles forming a certain compressive support structure.

The acidic water produced by the transformation of groundwater and clay minerals from dehydrogenation or decarboxylation of OM affects the dissolution of easily soluble minerals such as feldspar, quartz, and calcite, forming dissolution pores. The pore size of the dissolved pores in the particles is mainly distributed in the range of 0.05–4.00 μm, which is obviously smaller than the interparticle dissolved pores with a pore size of 1.00–4.00 μm (Figure 4i,j). There are common dissolution pores and fractures in the shale in the study area, and the dissolution minerals are mostly calcite and ferric dolomite. The intergranular pores of pyrite particles are the intercrystalline micropores formed by mineral crystallization under the conditions of a stable environment and appropriate reduction conditions, and the pore diameters are mostly distributed between 0.01 and 0.50 μm. The observation results of pyrite crystals in the shale of the study area show that the pyrite is mostly single crystals or irregular, and there are few pores between the pyrite grains. Most of the pyrite intergranular pores in the shale in the study area are filled with OM (Figure 4k,l).

### 3.5. Differences in Pore Structure Characteristics of Shale Reservoirs with Different Lithofacies

Different shale lithofacies have obvious differences in *TOC*, mineral compositions, and sedimentary structure, so different shale lithofacies also have different reservoir pore structures [12]. Organic pores are closely related to the *TOC* content. In the organic-rich lithofacies, organic pores are relatively developed, and the morphology of organic pores is complex (Figure 4a). The medium organic shale lithofacies contain fewer and elongated OM, and few organic hydrocarbon-generating pores exist (Figure 4b). Slit pores are mainly developed at the interface between OM and minerals. Organic-poor lithofacies contain sporadically distributed OM with rare OM pores developed. Different types and contents of clay minerals in different lithofacies result in differences in clay mineral pores. The organic-rich lamellar clay shale and the medium organic lamellar clay shale have the highest contents of clay minerals. The clay minerals are stacked in sheets or plates to form triangular pores. These pores are supported by clay minerals and have relatively high connectivity. Good OM filling is more common around the triangular pores (Figure 4d). There are polygonal pores formed by mutual support of clay minerals and quartz particles or pyrite particles in organic layered mixed shale and organic layered siliceous shale. The ore aggregates are mostly spherical and ellipsoid (Figure 4k). The organic-poor lamellar clay shale and the organic-poor massive mixed shale contain very little OM, and no pores are developed in the kerogen. The extruded pores of clay minerals are severely deformed and distorted, and most of the pores are extruded into slits. There are many dissolved pores in the organic-rich shale because the OM produces acidic fluids during thermal evolution and hydrocarbon generation, which causes the dissolution of brittle minerals, and migration of OM can also be seen in the dissolution pores. Dissolution pores can also be seen in layered mixed shale containing OM and layered siliceous shale containing OM. Due to the relatively small content of OM, dense dissolution pores are developed inside the calcareous shell and distributed along the shell. In the organic-poor lithofacies, due to the low content of OM, fewer acidic fluids are produced during the evolution process, so dissolution pores are rarely developed.

The spatial morphology and complexity of micro/nanopores in shale reservoirs can be inferred from the shape of nitrogen adsorption and desorption curves. In terms of pore morphology, the hysteresis loop openings of the organic-rich lamellar clay shale, medium organic lamellar clay shale, and organic-poor lamellar clay shale samples are relatively wide. The adsorption curve is gentle, the desorption curve is steep, and there is a large inflection point, which is in line with the characteristics of the H2-type hysteresis loop, indicating that there are more ink bottle-shaped pores and open-air permeability in this type of lithofacies (Figure 5a,b,e). The hysteresis loop of the samples containing organic layered siliceous shale and organic layered mixed shale was relatively narrow, the adsorption curve was generally flat, and the relative pressure rises sharply when the relative pressure was high. This result conforms to the characteristics of the H3-type hysteresis loop, but there are also certain inflection points, similar to the characteristics of the H2-type hysteresis loop (Figure 5c,d) The adsorption-desorption curves of N2 are shown in Figure 4. When the relative pressure (P/P_0_) is greater than 0.5, the isotherms of adsorption and desorption appear as hysteresis loops, indicating that capillary condensation occurs in mesoporous and macroporous ranges. According to the IUPAC classification, the type of hysteresis loop can reflect the pore morphology of porous materials [50]. This result reflected that this type of lithofacies mainly develops layered slit pores and has ink bottle pores. The hysteresis loop of the organic-poor massive mixed shale is very narrow, and the adsorption and desorption curves are close to horizontal, which is consistent with the characteristics of the H4-type hysteresis loop (Figure 5f). When the relative pressure is high, the adsorption and desorption curves also rise, which is consistent with the characteristics of the H3-type hysteresis loop, indicating that mesopores and micropores are simultaneously developed in this type of lithofacies, and there are also slit-type pores.

### 3.6. Wettability Characteristics of the Jurassic Continental Shale Reservoirs

Wettability is an important parameter of shale reservoirs and has an important impact on pore fluid distribution, capillary force, and permeability [51,52]. Factors such as OM content, mineral composition, and pore structure in shale reservoirs make the surface wettability of shale reservoirs particularly complex [53]. Through the analysis of the contact angle measurement results of different shale lithofacies samples, the overall contact angle of the shale in the study area is less than 45°, showing hydrophilicity (Figure 6). In the organic-rich rock facies, the contact angle is also relatively small, mainly because the clay minerals are hydrophilic. The higher the content of clay minerals, the stronger the hydrophilicity and the smaller the corresponding contact angle. The contact angle of the organic-rich rock facies is generally larger than that of the medium organic rock facies, which is mainly controlled by the OM content. The higher the OM content, the larger the contact angle. The degree of pore development on the surface of the sample is related. The layers with high carbonate mineral content in shale samples have poor pore development, which also leads to weaker hydrophilic ability and increased contact angle. Therefore, the heterogeneity of continental shale components in the study area also leads to a variety of pore structure types, which in turn leads to its overall complex wettability.

## 4. Discussion

### 4.1. Characteristics and Control Factors of Free Gas Volume

Natural gas in the free state in shale mainly exists in large matrix pores (>2 nm) in shale or in bedding or structural fractures in shale [54]. Similar to conventional natural gas, free gas has higher requirements on the pore and fracture space of the reservoir, and abundant free gas can be formed within large total pore volumes (porosity). The amount of free gas in the northeastern Sichuan Basin ranges from 0.11 to 6.3 m^3^/t, and the average free gas amount was 1.57 m^3^/t. The free gas content of the shale in the Da’anzhai Member was 0.23 to 3.84 m^3^/t, with an average content of 1.59 m^3^/t. The free gas content in the section was 0.11–6.3 m^3^/t, with an average content of 1.56 m^3^/t. Generally, the Da’anzhai member has higher free gas contents than the Dongyuemiao member. The amount of free gas in shale reservoirs was mainly controlled by the porosity and gas saturation of the shale reservoir. To further clarify the controlling factors of free gas in the continental shale of the Ziliujing Formation in the northeastern Sichuan Basin, the organic carbon content and mineral composition were considered. With the increase in the organic carbon content, the free gas content shows a certain increasing trend (Figure 7a). The effects of different brittle mineral contents on the free gas volume were compared. Quartz contents have a positive effect on the occurrence of free gas; high quartz mineral content is conducive to a large amount of free gas enrichment in the study area, but feldspar minerals, carbonate minerals, pyrite, and siderite show no obvious effects on free gas (Figure 7b–e). The probable reason may be the high heterogeneity of the continental shale reservoirs, and the various development of different brittle minerals in different regions and layers, which cannot provide a stable guarantee of free gas enrichment. The contents of different types of brittle minerals are summarized, and the relationship between the amount of free gas and these contents is analyzed (Figure 7f). It is found that the increase in the content of brittle minerals is beneficial to the occurrence of free gas. When the content of brittle minerals exceeds 60%, the amount of free gas can reach 1.5 m^3^/t, showing an excellent shale gas enrichment trend.

Different shale lithofacies with different pore structures and pore structure are key factors affecting the gas-bearing properties of shale reservoirs [55]. In general, the smaller the average pore size of shale, the more complex the pore structure, while the larger the pore-specific surface area, the stronger the adsorption capacity, and the higher the adsorbed gas content [13,56]. At the same time, the smaller the pore volume, the smaller the free gas content, and the more difficult the gas diffusion and seepage in the shale. When the average pore size of shale is larger, the free gas content is high, the adsorption capacity is also strong, and the pore uniformity is moderate, which is conducive to the seepage and development of shale gas (Figure 8). Therefore, there is a positive correlation between the free gas volume and the average pore size. The pore structure of shale mainly includes two aspects, pore size distribution and pore volumes, which directly affect the occurrence mode and reserves of shale gas and are important indicators for reservoir evaluation. The free gas volume of continental shale in the study area has a good positive correlation with the total pore volume, micropore volume, mesopore volume, and macropore volume. Under the same conditions, the larger the pore volume of shale, the greater the space for free gas to exist, and the greater the potential of its free gas content. Therefore, when the pore volume of shale increases, the amount of free gas stored also increases (Figure 9) [13]. Generally, when the pore size of shale is large, shale gas mainly occurs in the free state. The larger the pore volume, the higher the free gas content. Ross et al. (2009) [40] showed that when porosity increases from 0.5% to 4.2%, the free gas content in the shale will also increase from 5% to 50%. As one of the important parameters for calculating the free gas amount, the effect of shale porosity on the amount of free gas is crucial. In addition, continental shale is very sensitive to changes in water saturation in the reservoir, mainly due to the higher clay mineral composition of continental shale. Previous studies on the pore structure of continental shale in the northeastern Sichuan Basin under water-bearing conditions show that the most significant pore size range of continental shale pore structure changes with relative humidity (RH) is 2.5–20 nm. The pore volume of continental shale decreases significantly with increasing water saturation, and the maximum pore volume can be reduced to 1/2 of the dry condition [57]. The presence of water in the pores of shale leads to the expansion of clay minerals (especially montmorillonite), which greatly reduces the pore structure and reservoir properties of shale and thus greatly affects the free gas enrichment space.

### 4.2. Characteristics and Control Factors of Adsorbed Gas Volume

Adsorbed gas in shale refers to natural gas that exists in an adsorbed state on the surface of particles such as OM or clay minerals [58]. The earliest natural gas generated in shale reservoirs is quickly adsorbed on the surface or micropores of OM and various minerals. As specific surfaces and micropores are increasingly occupied by natural gas molecules, the adsorption rate will gradually decrease [59]. Natural gas dominated by physical adsorption on the surface of shale particles is easier to desorb. When conditions such as temperature and pressure change, natural gas in shale will have a dynamic equilibrium state of adsorption-desorption. The theoretical maximum gas adsorption capacity of continental shale in the Ziliujing Formation in the northeastern Sichuan Basin is 1.3–4.39 m^3^/t under dry conditions, with an average of 1.87 m^3^/t. Few samples have adsorbed gas above 3 m^3^/t. The Rankine pressure and experimental temperature in the isothermal adsorption test are low, while the pressure and temperature of shale in the actual underground reservoir are both high, and the underground reservoir has the influence of water molecules, so the amount of adsorbed natural gas in the underground reservoir should be slightly lower than the experimental test results.

There is an obvious positive correlation between the amount of adsorbed gas and the content of organic carbon (Figure 10a). As the content of total organic carbon increases, the corresponding amount of adsorbed gas also increases. Higher *TOC* contents lead to a larger gas generation potential and strong adsorption capacity, so the gas content per unit volume is high. Brittle minerals in shale mainly include quartz, potassium feldspar, plagioclase, calcite, dolomite, and pyrite. Brittle minerals usually have a weak adsorption capacity due to their small specific surface area. In this study, correlation analysis was carried out on the relationship between the amount of adsorbed gas and the contents of quartz, feldspar, carbonate rock, and pyrite. There is no obvious correlation between brittle minerals, and as the content of brittle minerals increases, the adsorption sites available for gas in shale even decrease.

Clay minerals occupy a large proportion of shale reservoirs, and their complex layered structure affects the adsorption of natural gas in shale reservoirs [60,61]. This is consistent with the research results of domestic and foreign researchers. The OM of continental shale is deposited with clay minerals, and the gas generated by OM is easily adsorbed in the micropores of clay minerals or on the surface of minerals. Therefore, when the content of clay minerals is high, the increase in micropores and specific surface area obviously increases the amount of gas adsorbed by shale. The clay minerals such as kaolinite, illite, chlorite, and illite mixed layers in the shale layer mostly exist in the form of flakes and plates, which are vertically stacked in space. Different components of clay minerals have different effects on the adsorption capacity. By establishing the correlation map between the amount of adsorbed gas and the mixed layers of kaolinite, chlorite, illite, and illite/smectite (Figure 11b–e), it can be seen that there is a weak positive correlation between the amount of adsorbed gas and kaolinite, chlorite, and illite minerals and a relatively obvious negative correlation trend with the mixed-bed minerals of illite. The increase in shale and illite minerals is beneficial to the adsorption of shale gas, while illite mixed-layer minerals are not conducive to the adsorption of shale gas.

During the deposition and burial process, the stratum will retain some water in its pore space. The shale layer is different from the conventional sandstone layer in that its pore space is small. It is rare in sandstone layers and mostly exists in the form of irreducible water [62]. However, this part of the water has a great influence on the amount of adsorbed gas in shale gas. Previous studies have found that the pore surface in shale has a certain effective adsorption site. The presence of water molecules will occupy these positions. For example, the intercrystalline pores of clay minerals that are abundant in shale layers are mostly negatively charged and are easily combined with water molecules, resulting in the inability of these pore surfaces to adsorb natural gas molecules, thereby reducing the amount of adsorbed gas. In the competitive adsorption process of water molecules and methane molecules, water molecules will preferentially adsorb on functional groups containing oxygen and nitrogen elements in kerogen, thereby reducing the amount of natural gas adsorbed [63,64]. In this study, the shale samples were processed by the equilibrium water experiment, and the adsorbed gas amount of the water-containing samples and the dry samples was obtained (Figure 12). By comparison, it can be seen that the adsorbed gas amount of the continental shale samples with different water contents in the Ziliujing Formation in the northeastern Sichuan Basin area is significantly lower than that of the dry samples. The effective adsorption sites of shale in this area are easily occupied by water molecules, which is closely related to the high clay mineral content.

The pore structure of shale also affects its adsorption capacity for methane [63]. The larger the specific surface area of continental shale, the more favorable the surface adsorption of methane. The larger the fractal dimension of the shale reservoir, the more complex the pore structure, and the easier it is for the adsorbed gas to be preserved and not easily lost (Figure 13). The adsorbed gas content has a strong correlation with the pore-specific surface area, showing that with the increase in the specific surface area, the amount of adsorbed gas in the shale increases gradually. This shows that the development of the pore-specific surface area in shale provides a large number of adsorption sites for adsorbed gas, which leads to the gradual enhancement of the adsorption capacity of shale with the increase in shale specific surface area, and the content of adsorbed gas is directly controlled by the pore ratio and surface area size. Wettability refers to the affinity of the rock surface to specific fluids, and the wettability of shale further affects the content of adsorbed gas. The surface of the substance with oil wetting generally has strong hydrogen bond attraction, weak lattice ion exchange capacity, or is an ionic compound with strong polarity, and this type of substance has excellent adsorption capacity for hydrocarbons containing hydrogen bonds. Hydrophilic minerals often interact with water to make the surface hydroxylated to form hydrophilic groups, and a large amount of water in the formation is preferentially adsorbed to occupy the space for hydrocarbon adsorption. External hydrophilic mineral crystals have a weak adsorption capacity for hydrocarbons, and it is difficult to form single-layer or even multilayer adsorption under strong molecular forces. Shale is composed of various components, among which quartz, chlorite, and illite are hydrophilic, calcite and feldspar are intermediate-wetting types, and OM, dolomite, and kaolinite are oleophilic. The amount of adsorbed gas in the shale in the study area has a good positive correlation with the water wetting angle, showing that with the increase in lipophilic substances, such as organic-rich laminar clay shale and medium organic layered clay shale, the amount of adsorbed gas also gradually increases (Figure 14).

### 4.3. Continental Shale Gas Enrichment Model

The high-steep structures in the northeastern Sichuan Basin belong to the “gap structure” in structural style, with high and steep anticlines and wide and gentle synclines. The mudstone caprock in the formation is thicker, with a cumulative thickness of up to 900 m, serving as a baffle layer. The fault throws are generally less than 200 m, and most of them do not break through the Jurassic, so the effect of faults on oil and gas enrichment is weak. Most of the shale gas distribution areas are located in the complex syncline area, with simple and stable structures, gentle strata, and underdeveloped faults [65,66]. The faults developed locally in the Middle-Lower Jurassic disappeared in the mud shale of the Qianfoya Formation and the gypsum-salt rocks of the Middle-Lower Triassic, and the structure and preservation conditions of the Da’anzhai Member are good [5]. During the depositional period of the Da’anzhai and Dongyuemiao members in Northeast China, the lake environment was favorable for the growth and development of plankton. OM can be enriched by a large amount of sedimentation under the background of a stable geological structure, and the fast subsidence rate can weaken the oxidation of OM to a certain extent, which is beneficial to the preservation of OM and forms a favorable facies belt for shale gas enrichment. In the center of the basin far from the mountain, due to the deep water body and the reducing sedimentary environment, the OM can be better preserved, and under the slow and long-term deposition conditions, thick shale formations can be formed, and the shale has good hydrocarbon generation. It is the most favorable enrichment facies belt for shale gas. In the wide and gentle slope area, a set of carbonate rocks are mainly developed. Under the influence of water bodies and climate change, some intervals can develop thin interbeds of mud shale, organic layered clay shale, and organic layered mixed layered shale. Lithofacies shale, such as shale, can also accumulate methane with relatively low OM content, but the gas content of shale gas is relatively lower than that of the deposition center. The enrichment model of the Lower Jurassic continental shale gas in the northeastern Sichuan Basin controls the material basis and high gas content in the center of the lake basin, the “baffle” layer controls the sealing and high pressure, and the complex syncline controls the preservation conditions, forming an enrichment pattern of the hyphenated-central baffle layer (Figure 15).

## 5. Conclusions

(1) The Jurassic continental shale in the northeastern Sichuan Basin has high clay content ranging from 33.3% to 67.4% and medium *TOC* content between 0.58% and 2.49%. The thermal maturity is in the mature-high maturity stage with the *R*_o_ in the range of 0.96–1.44%, and the kerogen types are mainly II_2_ and III.

(2) Six kinds of shale lithofacies including organic-rich lamellar clay shale, organic-rich layered clay shale, medium organic layered mixed shale, medium organic layered silicon shale, organic-poor lamellar clay shale and organic-poor massive mixed shale can be confirmed based on *TOC*, mineral contents, and sediment structures.

(3) The Jurassic Continental shales mainly develop organic pores, intergranular pores, pores between clay minerals, and pores between pyrite and dissolved pores. Affected by the OM content and material composition, the pore structure of different lithofacies reservoirs has obvious differences, and the overall contact angle of shale is less than 45°. The variety of pore structure types leads to its overall complex wettability.

(4) The amount of free gas in continental Jurassic shale is mainly controlled by OM content, brittleness, and total pore volumes, while the amount of adsorbed gas is mainly controlled by OM content, clay mineral type, wettability, and water saturation.

(5) The enrichment model of Lower Jurassic continental shale gas in northeast Sichuan is that the center of lacustrine basin controls the material basis, the “baffle” layer controls the sealing and high pressure, and the polysyncline controls the preservation conditions, forming the polysyncline and central baffle layer enrichment model.

## Figures and Tables

**Figure 1 nanomaterials-13-00779-f001:**
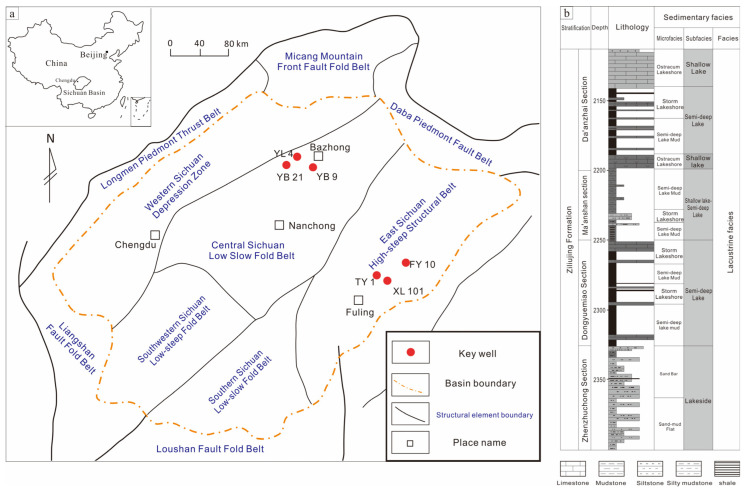
(**a**) Geological background; (**b**) stratigraphic sequence column map.

**Figure 2 nanomaterials-13-00779-f002:**
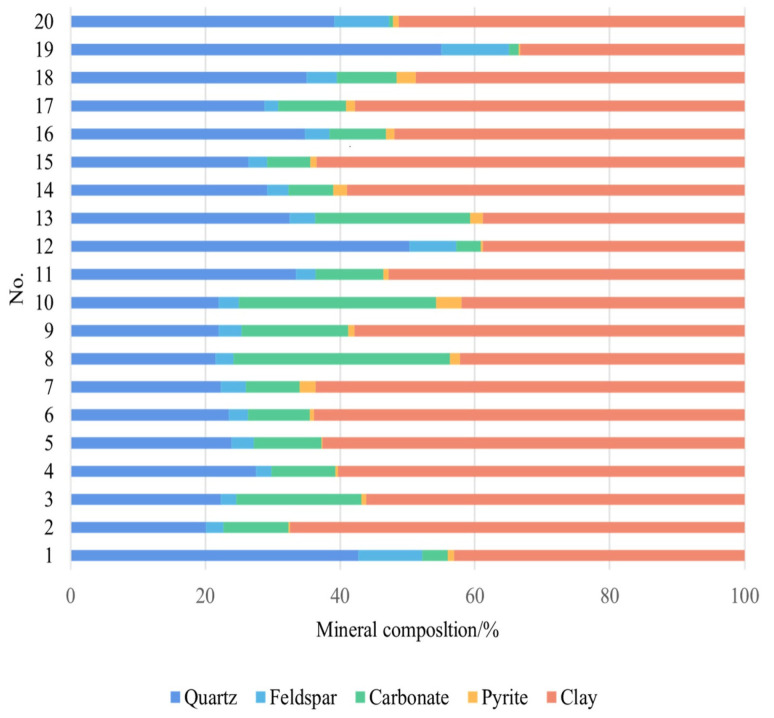
Mineral composition of continental shale in the Ziliujing Formation in northeastern Sichuan Basin.

**Figure 3 nanomaterials-13-00779-f003:**
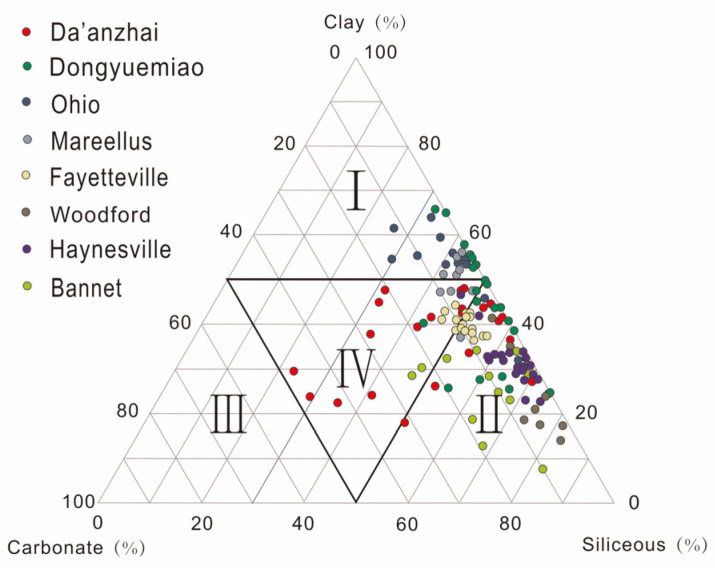
(I) clay shale facies; (II) siliceous shale facies; (Ⅲ) calcareous shale facies; (IV) mixed shale facies. Triangular diagram of mineral composition and lithofacies classification of continental shale in the Ziliujing Formation in the northeastern Sichuan Basin.

**Figure 4 nanomaterials-13-00779-f004:**
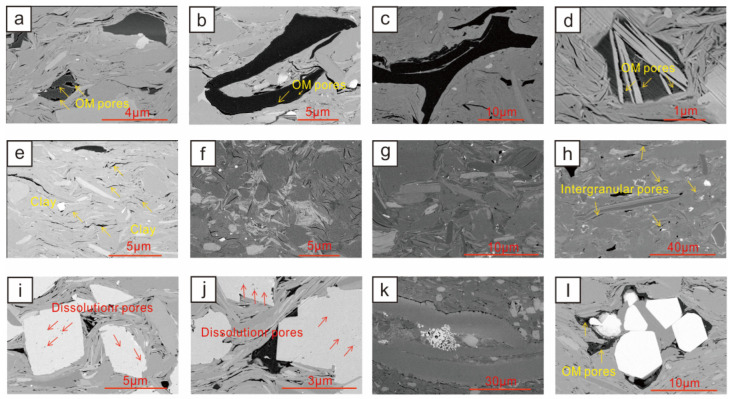
(**a**) YL4, Da’anzhai member, 3786.53 m, organic-poor massive silicon shale; (**b**) YL4, Da’anzhai member, 3760.75 m, medium organic layered mixed shale; (**c**) YL4, Dongyuemiao member, 3997.17 m, organic-rich lamellar clay shale; (**d**) YL4, Da’anzhai member, 3752.63 m, medium organic layered clay shale; (**e**) XL101, Da’anzhai member section, 2149.77 m, organic-poor lamellar mixed shale; (**f**) YL4, Dongyuemiao member, 4012.28 m, organic-poor massive clay shale; (**g**) YL4, Dongyuemiao member, 4051.81 m, organic-poor massive clay shale; (**h**) YL4, Dongyuemiao member, 3985.42 m, organic-poor massive mixed shale; (**i**) XL101, Dongyuemiao member, 2275.53 m, medium organic layered clay shale; (**j**) XL101, Dongyuemiao member, 2275.53 m, medium organic layered clay shale; (**k**) YL4, Da’anzhai member, 3748.23 m, medium organic massive clay shale; (**l**) XL101, Dongyuemiao member, 2269.94 m, medium organic layered mixed shale. FE-SEM image of continental shale pores of the Ziliujing Formation in the northeastern Sichuan Basin.

**Figure 5 nanomaterials-13-00779-f005:**
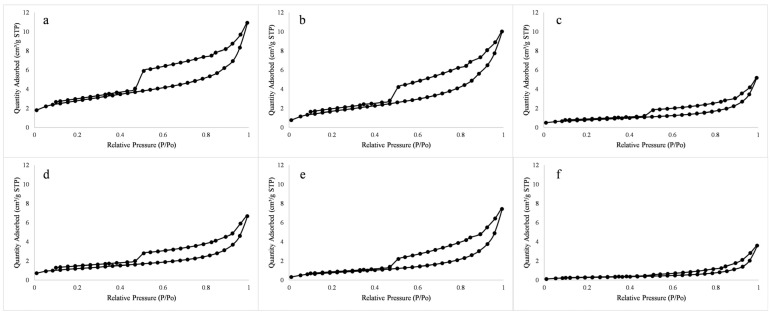
(**a**) Organic-rich lamellar clay shale (No. 20); (**b**) organic-rich lamellar clay shale (No.7); (**c**) medium organic layered mixed shale (No. 18); (**d**) medium organic layered silicon shale (No. 19); (**e**) organic-poor lamellar layered clay shale (No. 15); (**f**) organic-poor massive mixed shale (No. 1). Lines: Nitrogen adsorption loop curves; Pointes: Quantity adsorbed of different relative pressure. Nitrogen adsorption loop curves of shale with different lithofacies.

**Figure 6 nanomaterials-13-00779-f006:**
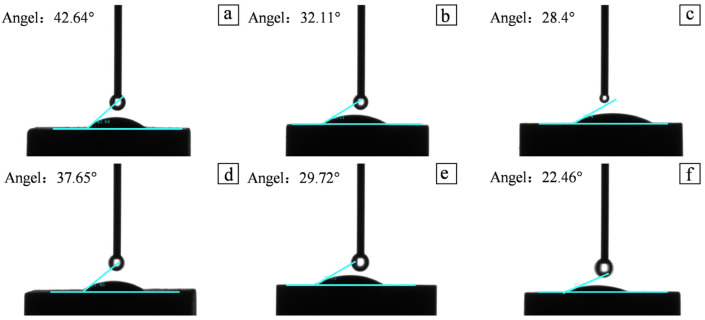
(**a**) Organic-rich lamellar clay shale (No. 20); (**b**) organic-rich lamellar clay shale (No.7); (**c**) medium organic layered mixed shale (No. 18) organic-rich lamellar mixed shale; (**d**) medium organic layered silicon shale (No. 19) organic-rich lamellar siliceous shale; (**e**) organic-poor lamellar layered clay shale (No. 15); (**f**) organic-poor massive mixed shale (No. 1). Determination of wetting angle of shale with different lithofacies.

**Figure 7 nanomaterials-13-00779-f007:**
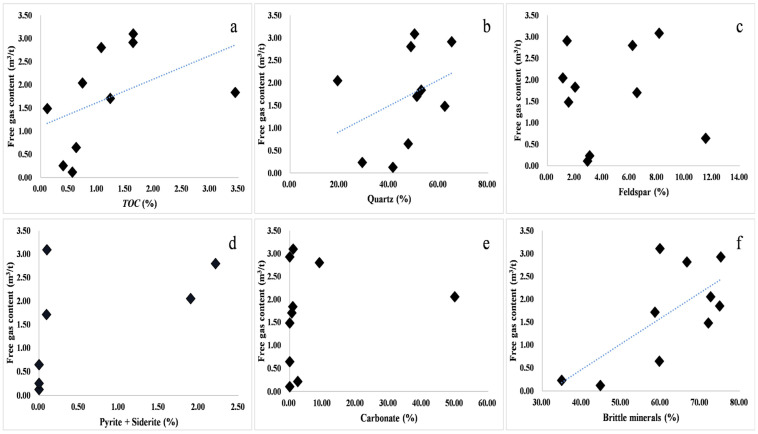
(**a**) *TOC*; (**b**) quartz; (**c**) feldspar; (**d**) siderite + pyrite; (**e**) carbonate mineral; (**f**) brittle minerals; Blue lines: Correlation trend line; Symbols: Sample point. Relationship between free gas content and *TOC* and brittle mineral content in the Ziliujing Formation shale in the northeastern Sichuan Basin.

**Figure 8 nanomaterials-13-00779-f008:**
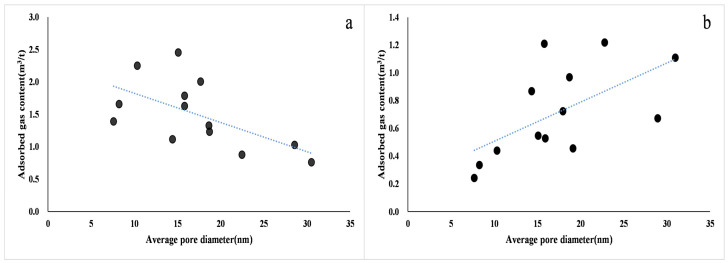
(**a**) Adsorbed gas; (**b**) Free gas; Blue lines: Correlation trend line; Symbols: Sample point. Relationship between gas content of Jurassic shale and average pore size of reservoirs in the northeastern Sichuan Basin.

**Figure 9 nanomaterials-13-00779-f009:**
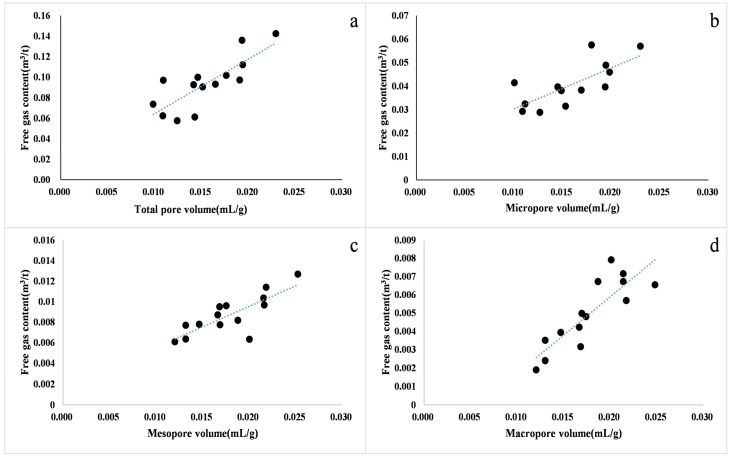
(**a**) Total pore volume; (**b**) micropore volume; (**c**) mesopore volume; (**d**) macropore volume; Blue lines: Correlation trend line; Symbols: Sample point. Correlation diagram of free gas volume and pore volume in continental shale reservoirs.

**Figure 10 nanomaterials-13-00779-f010:**
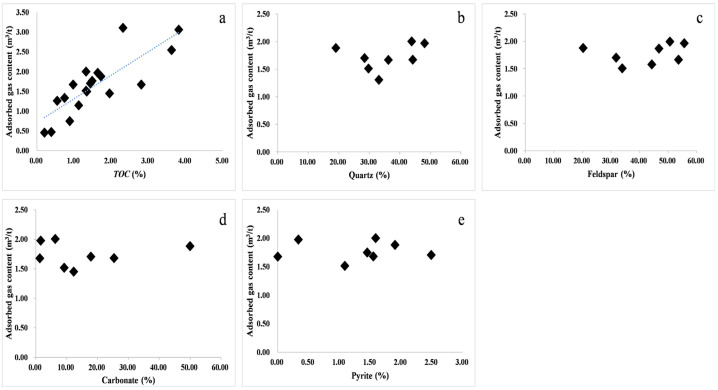
(**a**) *TOC*; (**b**) quartz; (**c**) feldspar; (**d**) carbonate minerals; (**e**) pyrite; Blue lines: Correlation trend line; Symbols: Sample point. The relationship between the maximum adsorbed gas volume and *TOC* and brittle mineral content in the Ziliujing Formation shale in the northeastern Sichuan Basin.

**Figure 11 nanomaterials-13-00779-f011:**
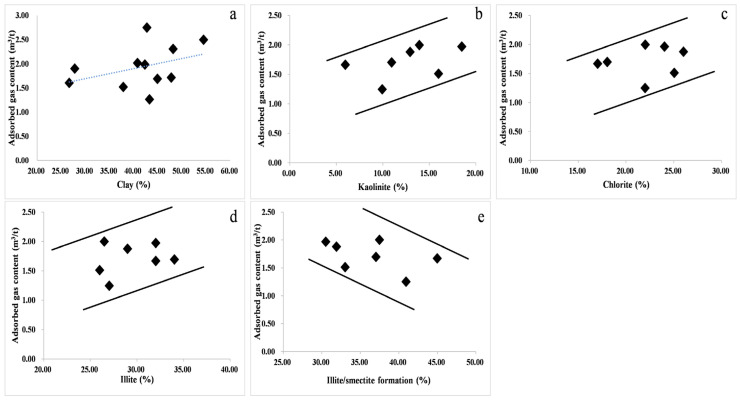
(**a**) total clay content; (**b**) kaolinite; (**c**) chlorite; (**d**) illite; (**e**) illite/smectite formation; Blue lines: Correlation trend line; Black line: Trend line; Symbols: Sample point. The relationship between gas adsorption and different clay minerals in the Ziliujing Formation shale in the northeastern Sichuan Basin.

**Figure 12 nanomaterials-13-00779-f012:**
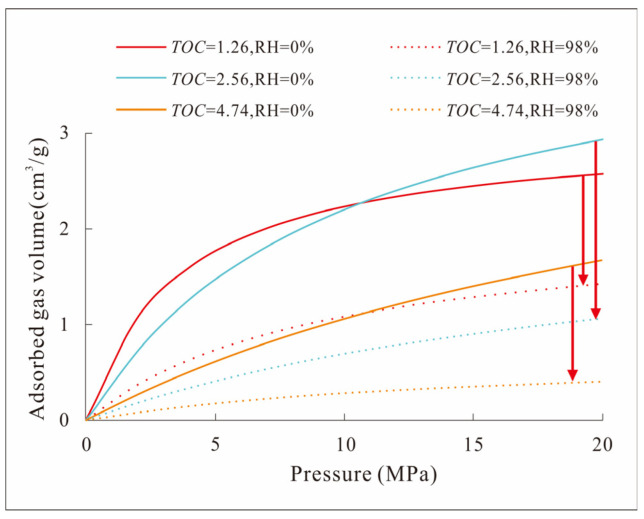
Arrow: The downward trend of adsorbed gas volume with the RH of different samples rising. Isotherm adsorption curves of the Ziliujing shale in the northeastern Sichuan Basin under dry and water-bearing conditions.

**Figure 13 nanomaterials-13-00779-f013:**
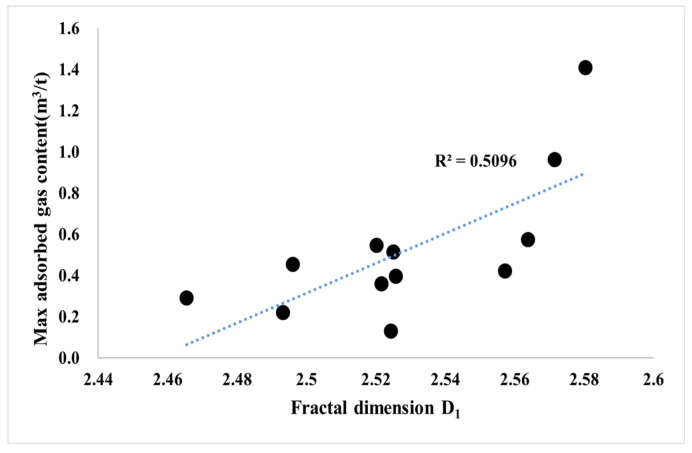
Blue lines: Correlation trend line; Symbols: Sample point. The relationship between the amount of adsorbed gas in continental shale and the integral shape dimension of the pore surface.

**Figure 14 nanomaterials-13-00779-f014:**
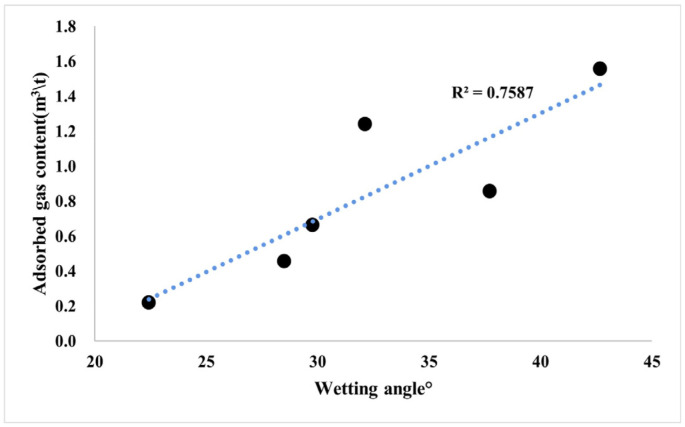
Blue lines: Correlation trend line; Symbols: Sample point. Relationship between gas adsorption and contact angle of the continental shale.

**Figure 15 nanomaterials-13-00779-f015:**
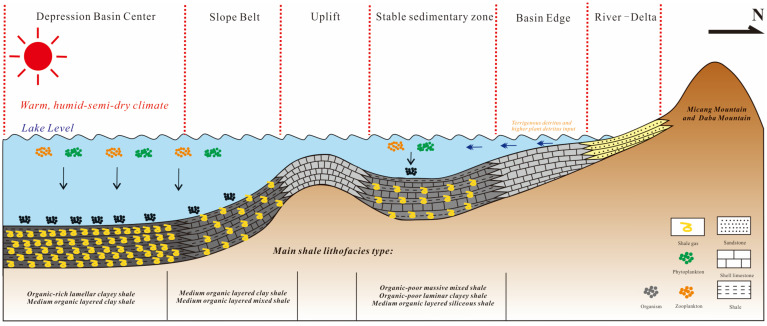
Shale gas enrichment pattern of the Ziliujing Formation in the northeastern Sichuan Basin.

**Table 1 nanomaterials-13-00779-t001:** Basic information of the Ziliujing well samples (Partial data from Jiang et al., 2021 [11]).

No.	Well	Depth/m	*TOC*/%	*R*_o_/%	Mineral Composltion/%				
					Quartz	Feldspar	Carbonate	Pyrite	Clay
1	XL101	2294.3	0.58	1.05	42.7	9.5	3.8	0.9	43.1
2	XL101	2275.6	1.97	0.96	20.1	2.6	9.6	0.3	67.4
3	XL101	2275.5	2.06	0.96	22.3	2.3	18.6	0.7	56.1
4	XL101	2274.2	1.84	1.02	27.5	2.3	9.5	0.4	60.3
5	XL101	2274.8	1.32	1.02	23.9	3.3	10.0	0.2	62.6
6	XL101	2269.7	1.46	0.99	23.5	2.8	9.2	0.6	63.9
7	XL101	2268.9	2.02	1.02	22.3	3.7	8.0	2.4	63.6
8	XL101	2158.2	1.74	0.96	21.5	2.7	32.1	1.5	42.2
9	XL101	2147.8	1.56	0.96	22.0	3.4	15.8	0.9	57.9
10	XL101	2144.6	0.77	0.96	22.0	3.0	29.2	3.8	42.0
11	YL4	3790.1	0.88	1.43	33.4	3.0	10.0	0.8	52.8
12	YL4	3786.5	0.70	1.44	50.3	6.9	3.7	0.3	38.8
13	YL4	3760.8	2.31	1.38	32.5	3.8	23.0	1.9	38.8
14	YL4	3755.5	0.79	1.36	29.2	3.1	6.7	2.0	59.0
15	YL4	3754.4	0.78	1.35	26.4	2.8	6.4	0.9	63.5
16	YL4	3752.6	1.33	1.31	34.8	3.6	8.4	1.3	51.9
17	YL4	3748.2	1.23	1.33	28.8	2.0	10.1	1.3	57.8
18	YL4	3735.46	1.16	1.31	35.0	4.5	8.9	2.8	48.8
19	YL4	3649.21	1.01	1.30	55.0	10.1	1.4	0.2	33.3
20	YL4	3646.36	2.49	1.30	39.1	8.2	0.5	0.9	51.3

## Data Availability

The data that support the findings of this study are available from the corresponding authors upon reasonable request.

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
