# Peer review of "Pore Structure and Gas Content Characteristics of Lower Jurassic Continental Shale Reservoirs in Northeast Sichuan, China"

_nanomaterials, 2023, doi:10.3390/nano13040779_

Round 1

Reviewer 1 Report

This is a very good paper providing new information on gas-in-shale potential of a sizeable sedimentary basin in southwestern China. I have only minor comments, which are summarized below.

1. Lines 45-46 - what about syn- and post-depositional tectonics? Surely, it will have some effect on gas distribution in sedimentary formations. Please add some references here.

2. Line 61 - any influence from diagenetic pyrite?

3. Figure 1 - For those of us not intimately familiar with Chinese geology, where exactly this area is located? Please add an inset map showing position of the Sichuan Basin in the grand scheme of the Chinese geology.

4. Line 98 - what peripheral orogenic belt? Does it have a proper name? Please specify.

5. Line 100 - "ancient continents" - I assume you mean cratons. Could you please specify and name them?

6. Phrase "organic-rich shale lithofacies" is duplicated in this sentence.

7. Section 3.3 - Addition of some thin section slides/pictures of different petrographic facies will assist readers to better understand and assess lithological features of these shale formations.

8. Figure 3 - authors need to supply scales for all SEM images. Also, additional comments directly on photos c, f, g and k will be very helpful in deciphering intricate porosity features of these sedimentary rocks.

9. Line 604 - Conclusion 4. I thought that the authors decided against any substantial influence from brittle minerals on absorbed gas volume in the Zilinjing Formation on the basis of graphs b, c, d and e in Figure 9. Could you please clarify?

Author Response

Thanks for editor-in-Chief and the reviewers. The reviewer's comments are in the attachment.

Reviewer 2 Report

The article contains the results of a study of the structure of a shale gas field to clarify and optimize the parameters of its development. With the help of well-known research methods, lithological facies are differentiated by structural features.

The presented classification of lithological facies and information on the porosity of the elements of gas-bearing rocks are of interest for designing the field development technology. Of scientific interest is also the specification of the concept of formation of continental shale gas in the studied part of the basin.

The in-depth information given in the article about the clay content of the deposit, as well as information about the various localization of free adsorbed gas, can become the basis for improving traditional technology with an environmental, economic and other effect. This also applies to the conclusion that free gas depends mainly on the content of organic matter, while adsorbed gas depends more on the type of clay and the saturation of shale with water. The authors believe that they have created the final model for the enrichment of continental shale gas, although there can be nothing final in science.

In our opinion, the sentence “Jurassic continental shale in the northeastern Sichuan Basin has a high clay content of (?)% and an average TOC content of 0.22% to 3.03%” omitted the value of clay content. The article contains original and novel elements. Its content is presented in an accessible way for the reader and illustrated enough to understand the text.

The significance of the content is determined by the specifics in assessing the state of the gas-bearing rock mass, which is important for the design of technology.

The scientific validity of the article follows from the representativeness of the cited sources.

The article will arouse interest among the readers to whom it is directed, specialists and teachers.

The article is of scientific value and can be published after editorial revision.

Author Response

(The authors gave the same response as above.)

Reviewer 3 Report

The manuscript titled “Pore structure and gas content characteristics of Lower Jurassic continental shale reservoirs in Northeast Sichuan, China” presents an interesting study that attempts to investigate the geological and geochemical characteristics of continental shale such as the pore structure and the gas-bearing properties, in order to define 1) the pore types of shale, 2) the influencing factors of gas-bearing properties.

The structure of the paper is the appropriate and has a logical order (Abstract, Introduction, Materials and Methods, Results, Discussion, Conclusions). The introduction is sufficient and informative. The results correspond to the conclusions. The literature has been used correctly.

However, there are a few minor corrections and clarifications that have to be made prior to publication in Nanomaterials Journal. Therefore, Minor Revision is suggested.

1. It is suggested to provide a figure with the mineral composition in sub-chapter 3.1.

2. Line 399 there is a possible misstatement, there is a full stop in the sentence, “..amount of free gas enrichment in the study area. feldspar minerals, carbonate..”.

3. Please provide information on how pore volume was determined.

4. Line 129, it is suggested to keep one form of expressing the figures among the manuscript either “Figure” or “Fig.”

5. Figures 4,5,7 need a better description in order to be more informative and sufficient for the reader, which samples they represent?

6. In figure 4 it is suggested to include in the description or in the diagrams the characterization of each hysteresis loop.

7. Figure 11 is suggested to be improved. Looks like it is cropped.

8.  Please provide more references in the discussion chapter, comparisons with the results of others are incomplete.

Author Response

(The authors gave the same response as above.)
